# Is Casual Sex Good for You? Casualness, Seriousness and Wellbeing in Intimate Relationships

**Aaron Ben-Ze'ev** 

Department of Philosophy, University of Haifa, Haifa 3498838, Israel; abenzeev@univ.haifa.ac.il

**Abstract:** Enduring romantic love is highly significant for our wellbeing, and there is much scientific evidence for its value. There is also evidence that marital sex is important for the flourishing of wellbeing for both partners. Casual sexual relationships and experiences (CSREs) are often characterized in a non-normative way, as sexual behavior occurring outside a committed romantic relationship. However, the prevailing normative description is negative, perceived as superficial behavior that harms our wellbeing. Although sexual activities are linked to many psychological and physical health benefits, these are rarely attributed to casual sex. Instead, scholars and laymen have warned against the negative consequences of non-committed sex, particularly for women. Yet, positive reactions to casual sex, such as satisfaction, confidence, self-knowledge and social engagement, are stronger and more common than negative reactions. Accordingly, the two major aims of this article are to understand the complexity of CSREs better, and to substantiate the claim that in various circumstances, CSREs contribute to our wellbeing.

**Keywords:** casual sex; romantic love; wellbeing; seriousness; hookup; sugaring



## Introduction

"You'll never meet my mom, strings will never be attached. We'll always get along, because it doesn't have to last". My Darkest Days, a Canadian Rock Band

Enduring romantic love is highly significant for our wellbeing, and there is much scientific evidence for its value. There is also evidence that marital sex is important for the flourishing of wellbeing for both partners. The definition of casual sex—behavior that occurs outside a committed romantic relationship—holds neither positive nor negative connotations. However, the prevailing normative description of casual sex is negative, perceived as superficial behavior that harms our wellbeing. Although sexual activities are linked to many psychological and physical health benefits, these are rarely attributed to casual sex. Instead, scholars and laymen have warned against the negative consequences of non-committed sex, particularly for women. Yet, positive reactions to casual sex, such as satisfaction, confidence, self-knowledge and social engagement, are stronger and more common than negative reactions [1]. Accordingly, the two major aims of this article are to understand the complexity of casual sex better and to substantiate the claim that, in various circumstances, casual sex contributes to our wellbeing.

I examine these issues through the conceptual construct of the "seriousness of intimate relationships", which refers to both romantic and sexual relationships. I focus on four major factors underlying seriousness: temporality, profundity, commitment and authenticity. A common assumption considers seriousness to correlate with our wellbeing in a linear way. Rejecting this assumption is not due to the notion that highly serious intimate relationships often create a high quality of life, but rather to the claim that those who are least serious necessarily have a low quality of life. I argue that intimate relationships, which have a low degree of seriousness, such as those involved in casual sex, may enjoy benefits regarding our wellbeing that those higher on the seriousness scale may not have. The extent of positivity is not necessarily correlated with the degree of seriousness. Accordingly, my

main claim is that, in many life circumstances, casual sex can contribute greatly to our wellbeing. These circumstances are not without limits, but after they are established, casual sex can enhance the flourishing of each partner. My attitude is Aristotelian: in certain balanced circumstances, casual sex may be good for you, yet finding the optimal balance is often not straightforward.

I begin by briefly mentioning the relationships between love and sex and then discuss the notion of seriousness. I then examine various types of casual sex, as well as types of commercial casual sex, focusing on the increasingly common phenomena of hookups and sugaring. The last sections examine the benefits of casual sex and the need for a balanced diet.

## 1. Casual Relationships and Seriousness

"Angels fly because they take themselves lightly". G. K. Chesterton, English writer

Casual sexual relationships and experiences come in various forms. In order to examine their connection to our wellbeing (in the sense of the quality of life and the good life), I examine the issue of seriousness in intimate relationships. I begin by briefly describing the connection between love, sex and wellbeing.

### 1.1. Love, Sex and Wellbeing

"My marriage is pretty great. But I think about other guys all the time". A married woman

Love is considered one of the most meaningful and sublime human expressions. Conversely, sex is often seen to involve vulgar and humiliating activities that degrade the partner into a commodity [2] (Chap. 14). Accordingly, people are often insulted when believing that their partner's love is based merely on sexual desire, but they are similarly insulted when realizing that their romantic partner has no sexual desire for them. However, sexual desire is typically central in romantic love and the two overlap a great deal in the brain, activating specific, related areas [3]. Nevertheless, sometimes, sexual desire has nothing to do with romantic love.

The precise nature of wellbeing is disputable. I adopt a common general sense, indicating that wellbeing constitutes optimal psychological functioning and experiences. Various studies indicate that good romantic relationships are of great value in maintaining our wellbeing by being closely linked to personal happiness, associated with higher rates of self-esteem, safety, life satisfaction and positive outcomes and achievement of personal and relationship goals. However, romantic relationships have also been associated with negative outcomes [4,5].

There is much evidence for the advantages of sex in romantic relationships, including promoting wellbeing and having long-term relational benefits [6]. Although sexual urges and emotional attachments are not always connected, joint operation of the sexual and the attachment systems is, nevertheless, typical of romantic relationships [7]. Women's sexual satisfaction was found to be an extremely strong predictor of relationship wellbeing, a strong predictor of mental health and a weak to moderately strong predictor of physical health [8]. Likewise, Natasha McKeever [9] suggested the following positive outcomes of romantic love that involves sex: (i) pleasure, (ii) union/physical closeness, (iii) intimacy and (iv) vulnerability and care.

Sexual satisfaction within marriage (and other committed relationships) is highly correlated with relationship satisfaction. However, my major concern here is not the value of sex within committed relationships, but rather the value of casual sex, which is characterized as sex outside an ongoing committed relationship. The above findings are often perceived to oppose the value of casual sex. Contrary to this, I believe that casual sex has its own intrinsic value.

Much attention has been given to the morality of sex. Immanuel Kant famously argued that sex "makes of the loved person an object of appetite. Taken by itself, it is a degradation of human nature" [10] ( p. 163). However, the morality of sex is considerably more complex (see, e.g., [11,12] (pp. 200–210); [13–17]). Alan Soble [18] discusses the difference between

the moral and nonmoral aspects in sex and describes four major combinations. A sexual act might be *both morally and nonmorally good*, as in the case of the exciting sexual activity of a newly married couple. A sexual act might be *morally good and nonmorally bad*, as in the case of the routine sexual act of a couple after ten years of marriage. A sexual act might be *morally bad yet nonmorally good*, when one partner commits adultery. Further, a sexual act might be *both morally and nonmorally bad*, when the adulterous couple get tired of each other, unable to experience the excitement they once shared [19]. In this classification, nonmorally good sex is expressed in being satisfactory and enjoyable.

My interest here is more limited, since it focuses on casual sex, rather than sex in general. However, this interest is not limited to the satisfactory and enjoyable nature of casual sex, but rather whether casual sex contributes to our wellbeing. I claim that casual sex has its own benefits for our wellbeing and flourishing.

### 1.2. Seriousness

"Life is too important to be taken seriously". Oscar Wilde, Irish poet and playwright

Seriousness is commonly perceived as an important virtue, involving other characteristics, such as being earnest, sincere and showing deep thought. However, seriousness is only one valuable trait and being casual and light, while having a sense of humor, is equally important. Hence, excessive seriousness, which leaves no room for the casual and light side of life, may also be criticized. Jean-Paul Sartre accordingly called to "repudiate the *spirit of seriousness*" of traditional philosophy, as well as of bourgeois culture [20] (p. 796).

The term "serious" has various meanings. The *Oxford Learner's Dictionaries* indicate the following major meanings: not silly ("thinking about things in a careful and sensible way"), sincere ("not joking"), needing thought ("needing to be thought about carefully and not only for pleasure") and important ("must be treated as important"). These meanings adequately express the differences between casual sex and enduring romantic relationships. Thus, in comparison to enduring romantic relationships, casual sex is carried out solely for pleasure, treated by those involved as less important, regarded as sillier and less sincere and involving less thought. An additional dictionary normative meaning to "serious" is bad or dangerous. It is surprising that such a negative description is ascribed to experiences that are regarded as overwhelmingly positive. We may explain this negative attitude as a result of the risks of being too serious.

Psychologists occasionally claim that seriousness is the opposite of casual activities [21]). The concept is also used to differentiate between various activities, where, for example, serious leisure activities are characterized by their perseverance through difficulty, an intention towards continued involvement, as well as an effort to acquire skills and knowledge of the chosen leisure activity. Low levels of seriousness reflect a casual approach to leisure that entails engaging in activities for short-lived pleasure or relaxation [22,23].

### 1.3. Factors Underlying the Seriousness of Intimate Relationships

Intimate relationships can be serious to various degrees. I focus, here, on four major factors underlying the degree of seriousness of intimate relationships: (a) temporality, (b) profundity, (c) commitment and (d) authenticity.

#### 1.3.1. Temporality

We can talk about three major temporal types of emotional experiences: (1) acute emotions, (2) extended emotions and (3) enduring emotions. *Acute emotions* are brief, almost instantaneous experiences. *Extended emotions* involve successive repetitions of experiences that are felt to belong to the same emotion—for example, being angry or jealous for hours. *Enduring emotions* are the longest lasting and can persist for a lifetime. Romantic relationships involve all types of emotional experiences: acute, extended and enduring love. Sexual encounters involve acute, and sometimes extended, sexual desire [24].

Affective time has various aspects; here, I focus on duration, continuity, frequency and meaningful direction. *Duration* concerns the length of the relationship (or encounter);

*continuity* refers to the amount of awake time that couples spend together (companionship). *Frequency* refers to its repetition—the rate at which the experience reoccurs. *Meaningful direction* refers to the development (or deterioration) of a relationship experience over time. These aspects are relevant to the distinction between romantic relationships and casual sex, and the distinction between various types of casual sex.

Both duration and continuity are necessary, but not sufficient, for developing serious romantic relationships. Thus, being together for a longer time does not necessarily mean loving more deeply. Similarly, spending time together increases the couple's romantic profundity, but spending too much time together, without sufficient personal space, can damage the relationship. In any case, continuously not paying attention to one's partner while being together is damaging. Thus, a study of phubbing (ignoring a conversation going on around oneself to focus on one's mobile phone) indicates the value of continuity by revealing that phubbing negatively impacts relationships and life satisfaction [25].

In casual sex, duration can be a matter of minutes or hours, and not of months or years. Nevertheless, what is most significant in the quality of sexual encounters is the duration and continuity of the whole sexual experience, from foreplay through the sexual act itself to its afterglow. Sexual afterglow, the good feeling that lingers after pleasurable sexual experiences, is particularly significant. Research suggests that afterglow, rather than orgasm, determines how people feel about their sexual partner. Spouses who have experienced stronger afterglows report higher levels of marital satisfaction, both in the moment and over time, compared to other spouses [26]. Romantic partners view the time after intercourse as important for bonding and intimacy. Frequent physical affection, such as kissing, cuddling and hugging, was found to significantly increase the duration and the quality of the relationship. Thus, one study of newlywed couples showed that sexual afterglow remains for approximately 48 h after sex, and those with stronger afterglow had higher overall marital satisfaction [27–29].

In the acute emotion of romantic love, frequency may indicate the intensity of the relationship and the emotions associated with it. In casual sex, which mainly involves acute and extended sexual experiences, the issue of frequency is at the center. Especially in the case of men, rather than women, it is common to brag about the number and frequency of sexual encounters as well as the number of orgasms in each [30,31](. In addition to their duration, continuity and frequency, enduring romantic relationships involve a qualitative meaningful direction of development (or deterioration) and a dispositional nature that unfolds over time. In this sense, we can distinguish between external change and intrinsic development. *Change* is commonly taken to mean becoming different, typically without permanently losing one's characteristics or essence. In its positive sense, used here, *development* is a specific type of change that involves steady growth, improved through expanding or refining. Such development, which requires the time, and sometimes the effort of getting to know another's unique personality and circumstances, is meaningful and is a kind of achievement [24].

### 1.3.2. Profundity

The above temporal differences relate to the second factor of seriousness of intimate relationships, namely, the superficiality and profundity continuum. Compared to casual sex, romantic relationships are more profound by far.

Something that is profound extends far below the surface and has lasting effects. Profound emotional experiences have a lingering impact on our life and personality. Profound activities, which are sometimes unpleasant, typically involve deep, meaningful satisfaction in overcoming difficulties while using one's most distinctive capacities [24] (pp. 41–42).

In the romantic realm, we can distinguish between erotic intensity, which is a snapshot of a momentary peak of passionate, often sexual, desire, and romantic profundity, which extends beyond mere intensity and refers to the lover's broader and more enduring attitude. External change is highly significant in generating erotic intensity, yet in romantic depth, familiarity, stability and development are more important. Romantic profundity involves certain types of activities that evolve over time and, as a result, are often absent in casual sex.

While romantic novelty is useful in *preventing* boredom, romantic familiarity is valuable in *promoting* flourishing. Erotic intensity characterizes casual sex and early stages of romantic relationships, whereas profundity takes place during the latter stages of romantic relationships [13,24].

Profundity and intensity are not different kinds of romantic love, but rather different features present in various degrees within different romantic and sexual relationships. There is often a reverse correlation between intensity and profundity. Intensity is typically high at the beginning of a relationship, while profundity is high at the more mature stage, though low at the very beginning. Nevertheless, there are romantic relationships in which both intensity and profundity are at a moderate level, or cases in which they are both very high or very low.

### 1.3.3. Commitment

The tension between stable boundaries, which secures our comfort zones, and the wish to experience novelty, which is often produced by transcending those boundaries, is basic to human life and the experience of love. This is also a tension between the ideal of freedom, which is often expressed in casual sex, and the ideal of commitment that is essential to enduring romantic relationships.

There is no romantic life without a sense of meaningful belonging [32]. Belonging comes at a price: it limits the number of romantic partners we can have—after all, belongingness involves commitments and the allocation of scarce resources. The strength of our commitment depends on various personal and circumstantial factors. Thus, while the presence of quality relationship alternatives decreases romantic commitment, expected future satisfaction with the relationship enhances commitment, even more so than our current satisfaction with that relationship [33].( In some circumstances, commitment to another person may reduce your own authenticity.

Romantic commitment does not disintegrate without cause—there should be good reasons to breach a romantic commitment. Such commitment mainly stems from our relationship with our partner and not from comparing the partner to other people. Shared history is highly relevant to the issue of commitment, which is enhanced over time. Our commitment to someone we have been with for ten years is far greater than to the one we are with for merely one night. It is evident that lovers have some commitment toward their beloveds and that this makes the transfer of love from one person to another hard [34,35]. Commitment should be respected, but not at any price; excitement, development, diversity and complexity should also be appreciated, but again, not at any price.

In short-term relationships, where commitment is low, people prefer dissimilar partners. In long-term relationships, which are characterized by high levels of commitment and joint activities, greater similarity correlates with greater romantic value. In serious romantic relationships, commitment and satisfaction increase over time [36–39].

### 1.3.4. Authenticity

Authenticity, the quality of being genuine or real, refers to the degree that one's actions and perceptions are congruent with one's values and desires. Authenticity is regarded by many philosophers and psychologists as vital to our wellbeing in general, and to serious romantic relationships in particular. However, the nature of authenticity is complex. Ronald De Sousa argues that authenticity, which is acting truth to oneself, demands the depth dimension of counterfactual possibility and differentiates us from one another rather than leading us along the same path [40]. Authenticity is related to another key characteristic in romantic relationships, namely, autonomy. Autonomy, which emphasizes the individual's self-governing abilities, enables us to behave in an authentic manner. However, the two are not identical. Authenticity demands more than is necessary for autonomy: a person does not have to endorse key aspects of their identity in order to qualify as autonomous [41].

There is no doubt that in enduring, profound committed romantic relationships, authenticity is significant. Authenticity is less significant in superficial relationships. As we shall see

below, determining authenticity is considerably more complex because of the personal nature of authenticity and the presence of various degrees and types of authenticity.

Take, for example, commercial casual sex. Elisabeth Bernstein coined the term "bounded authenticity", which refers to the common sale and purchase of authentic emotional and physical connection. This is contrary to the quick, impersonal sexual release associated with street-level prostitution. Bernstein found that sex workers try to manufacture authenticity by trying to simulate, or even produce, genuine desire, pleasure and erotic interest for their clients, while endowing them with a sense of desirability, esteem or even love. While doing so, they also create a meaningful intimate experience for themselves. Bernstein reports a sex worker who really enjoyed having sex with an attractive man—a rare thing—and who offered him a special, lower price for their encounters. For another man she was attracted to, she suggested he might "come for free". Both men panicked and never returned since they wanted an emotional connection without intimacy and obligation [42]. Bounded authenticity has some degree of seriousness, which some people, both clients and sex workers, may not want to have.

To sum up, I have examined the notion of serious intimate relationships by focusing on specific factors that are relevant to whether casual sex can contribute to our wellbeing: (a) temporality, (b) profundity, (c) commitment and (d) authenticity. In the first three factors, casual sex is typically less serious than romantic relationships. In comparison to romantic relationships, casual sexual relationships and experiences are briefer, more superficial and less committed. As we shall see below, comparing authenticity in casual sex to authenticity in romantic relationships is more complex. In any case, the above discussion does not imply that casual sex does not contribute to our wellbeing, for it certainly does.

## 2. Casual Sex

"My first one-night stand turned into a three-year relationship". A woman

In this section, I discuss various types of casual sex. Understanding the differences between these types is useful for determining the contribution of casual sex to our wellbeing.

### 2.1. Types of Casual Sex

Jocelyn Wentland and Elke Reissing divide casual sexual relationships into four major types: "one-night stands", "booty calls", "fuck buddies" and "friends with benefits". One-night stands, which are the briefest sexual relationships, are often regarded as the most superficial encounter, typically taking place between strangers or after brief acquaintance. Booty calls refer to a communication initiated with the urgent intent of having a sexual encounter. Fuck buddies are already friends, but their friendship is largely limited to sexual interactions. Friendship with benefits involves the most profound activity of casual sex, where the partners are first friends and then they add the bonus of sexual benefit [43].

### 2.2. One-Night Stands

"I think every woman should have a one-night stand. If it's done right, it can be liberating". Rachel Perry, Canadian TV personality

One-night stands are often regarded as the most superficial and least serious form of casual sex; accordingly, one-night stands are perceived to be the opposite of serious love. It, therefore, appears counterintuitive that they will develop into long-term, serious romantic relationships. The very term, "one-night stand", indicates that it is a brief, superficial and inconsequential experience, taking place only once, for the specific purpose of sexual gratification.

Contrary to this reasoning, Helen Fisher [44] found that about 27% of those who had one-night stands testified that their experiences turned into committed long-term relationships. Fisher argues, in a paper titled, "Casual sex may be improving America's marriages", that in learning a lot about the person between the sheets, it is possible to kick-start a real relationship. Moreover, another study found that only 15% of friends with benefits lead to a committed long-term relationship [45]. Furthermore, The Knot 2021 study

of dating apps shows that although Tinder has a reputation for generating mainly casual sexual relationships, and in particular one-night stands, Tinder was responsible for pairing about a quarter of newlyweds who met online, making it the best dating app for marriage. Another surprising finding is that individuals were more likely to leave immediately after sex in the context of booty-call relationships than after one-night stands [46,47]. This is contrary to the assumption that one-night stands are less serious than booty calls.

A major explanation for these surprising results may refer to expectations of one-night stands; they are intended to be the first and last time the two meet. Accordingly, participants may feel greater freedom and openness to be themselves and behave in an authentic manner, without worrying about their lover's reactions. This increases self-disclosure, which, in turn, enhances intimacy. The very limited time of one-night stands provides a kind of secure environment, enabling a more authentic behavior. In a similar manner, a greater sense of security in online relationships, due to physical separation, increases self-disclosure and disinhibition in these relationships [48]. In a sense, a one-night stand is an isolated authentic island, separate from the difficult and complex reality around it. This is its strength, as well as its weakness.

### 2.3. Booty Calls

Unlike one-night stands, which are extended sexual encounters, booty calls exist as an ongoing relationship, engaging in repeated sexual activities with someone familiar. Despite the acquaintance, booty calls are not planned, participants do not consider each other friends, they typically do not stay overnight and they share minimal affection. The unpredictability and spontaneity of booty calls are some of their valued characteristics. When booty calls become too regular or frequent, the participants are considered to be fuck buddies.

Peter Jonason and colleagues [46,47] showed that although booty-call relationships often lack romantic acts found in serious, long-term relationships (such as talking and handholding), more erotic intimate acts, such as kissing, manual sex, fondling of breasts/the chest and anal sex, were found to occur more often in booty-call relationships than in one-night stands.

Evita March and colleagues examined the types of emotional and sexual acts involved in booty-call relationships and compared their frequency to one-night stands and serious long-term relationships. They found that booty calls tend to align more strongly with short-term, rather than long-term, relationships, since they are relatively sexual in nature, and while physical attractiveness is a highly valued trait, commitment is lacking. Demonstrative of the sexual nature of booty-call relationships, the individual tends to leave after sex and not engage in handholding. In contrast, the romantic nature of booty-call relationships was demonstrated through the frequency of acts such as kissing. This indicates that a booty-call relationship may be a hybrid of a long- and short-term relationship, constituting a compromise that allows men to have sex without a high level of commitment, while offering women the potential for future commitment [49].

### 2.4. Fuck Buddies

"I have had sex with my lover a few times at a hotel; it means nothing to me. It's like drinking water and not being quenched. My body is connected with him, but my soul does not. I implied to him that I'd rather be a fuck buddy, rather than more. Trying to keep the emotional attachment at arm's length". A married woman

A fuck buddy is a sex partner with whom one has sex, usually repeatedly, but without any romantic attachment and no strings attached. The two people simply enjoy having sex with each other. This does not mean that you do not ever talk with your fuck buddy, but you do not spend time together unless sex takes place. The two buddies do not have much in common, and there is often a significant personality or age difference. Neither really wants a serious relationship at the time but they both typically have high sex drives.

A major difference between booty calls and fuck buddies is temporal: whereas booty calls take place in an irregular manner (only when there is the need to have sex), fuck buddies regularly meet, unless there is something of greater importance that prevents it. Buddies should explain why they cannot meet, where booty-call partners do not need to provide such information. In this sense, fuck buddies lean closer to romantic relationships and have a higher level of seriousness.

In an article in *Cosmopolitan* (10 January 2020), titled "How I had a successful fuck buddy situation for two years", the journalist Jasmine Lee-Zogbessou describes her own experience: "A few years ago, I began sleeping with a friend. He became my fuck buddy, on and off, for two years. I didn't get attached, I never lost sleep over what we were, and I never tried to have a serious conversation about where we were heading. I happily explored sex like never before. Having a fuck buddy became something I never knew I needed". Communication was a large part of the reason why Jasmine and her fuck buddy were both always satisfied. This relationship helped her understand that not everyone was a potential boyfriend, and a casual relationship could be more beneficial than a serious one. Casual sex should only be for those who authentically desire casual sex, not for those who think it is all they can ask for from someone else.

The features associated with fuck buddies, such as limited expectations, seeing casual sex as having an intrinsic value and completely open communication, are what make this relationship most valuable in various circumstances.

### 2.5. Friends with Benefits

"No man can be friend with a woman that he finds attractive. He always wants to have sex with her". Harry, in the movie, *When Harry Met Sally*

I turn now to discuss Friendship with Benefits (FWB), which is an increasingly common type of casual relationship and the closest in its seriousness to romantic relationships. FWB involves the two major aspects of romantic love: friendship and sex. However, it differs from romantic relationships in its lesser seriousness expressed in its temporality (it is briefer, without continuity, and less frequent), in being less profound and requiring less commitment.

FWB constitutes an intermediate, unstable and relatively brief experience. Yet, such a friendship can last weeks and months, extending to several years. Laura Machia and colleagues [45] found that FWB is indeed relatively short: about one-third of the participants in the study reported that their relationships did not survive the first year, and those that did turned back into regular friendships. In contrast, the majority of those who wanted to transition into romantic relationships did not do so. Once the option of development into a romantic relationship evaporated, the combination of a friendship and a sexual partner ceased to have value for them.

Given the restless nature of our world, the relatively longer duration of FWB, compared with other forms of casual sex, is also valuable. Unlike marriage, FWB does not prevent its participants from looking around and finding another, more fulfilling relationship. FWB is a compromise in which one gives up romantic profundity and manages with being second best. This compromise can be valuable and enjoyable. In economic terms, FWB cuts the costs and reduces the revenue. It cuts an emotional cost, since there is a minimal price to pay and the relationship is relatively risk-free, unless it ruins the friendship itself. The revenue is reduced because enduring profound romantic love is excluded. FWB is not suitable for all people or for every period of our lives. It is particularly difficult when the friends are married and have young children.

### 2.6. The Hookup Culture

Hookup culture is not another type of casual sex but rather a general behavioral framework that includes various specific types of casual sex, such as one-night stands and booty calls. In hookup culture, which prevails among many young people, especially in colleges, casualness is a culture suited to the circumstances of being young, having many sexual opportunities and not yet ready to commit. Lisa Wade [50,51] indicates four major rules for

maintaining casualness in hookup culture: drunkenness, aloofness and avoiding tenderness and repetition. Drunkenness functions symbolically to code sexual activity as non-romantic. Aloofness demonstrates that an encounter does not involve an authentic emotional bond. Avoiding tenderness indicates a lack of care, which is essential in serious romantic relationships, and rejecting the repetition of hookups with the same person ensures that people do not start an intimate relationship.

We should distinguish between this extreme hookup culture and the hookups themselves. Although this extreme culture is common, there are many hookups that are closer to typical casual sex, which involves some degree of seriousness. One study that explored the benefits of hooking up among first-year college women found benefits closely related to their wellbeing, such as sexual satisfaction, general positive emotions, increased confidence and clarification of feelings [1,52,53].

Considering the four major factors of seriousness in intimate relationships analyzed here, hookup culture expresses an extreme version, which sits very low on the scale of such seriousness and is contrary to the romantic ideology in which love is intended to be profound, committed, authentic and endure forever [54]. In all temporal aspects discussed here, hookups feature the lowest degree of seriousness—the *duration* is very brief, without *continuity* and *frequency*, and they contain no *meaningful development*. In this way, hookups express an extremely superficial attitude, an expression of a shallow culture. Likewise, with the extreme implementation of "no strings attached" and the ideal of creating emotional distance, there is no commitment whatsoever, not even sending polite messages after the hookup. This leads to establishing, in hookup culture, the common advice of being less close after a hookup than before [50].

As in other extreme beliefs, hookup culture has difficulties in avoiding the "danger" of serious relationships. Being drunk with no use of reasoning is necessary for exercising extreme casualness. As one student said, "not being drunk is unnatural and abnormal".

There are cases of bullying students who choose not to drink, since being sober involves a real choice, and things are "getting real". Hookups are praised for providing the greatest sexual freedom for participants, but this is far from the case. As Wade cites one student, "many hookups are dictated by how our peers view the potential partner ... I am unable to separate my opinion from those of my friends". In this case, attractiveness is in the eyes of the beholders, plural [50].

I believe that freedom does not express a lack of values but rather corresponds with acting authentically accordingly to our profound values. Extreme hookup culture may be of some value in specific limited circumstances but not as a guide to real intimate behavior. The difference between the extreme form of hookups and traditional casual sex is that in the latter, there is no attitude that forbids development of serious intimate relationships—on the contrary, it is often an initial stage on the road to serious relationships. The main problem of this form of hookup culture is that it is too serious about casual sex.

I would also like to mention that for most singles, casualness is not a lifetime attitude but a specific attitude toward connecting with a certain person. Indeed, a Match survey of singles in the USA (2022) [55] shows that 80% of young singles indicate that they would like to find long-term relationships. Casualness for these singles is expressed in the fact that they do not want to give up their sexual freedom for the sake of a significant serious relationship. Hence, alongside their search for serious romantic relationships, singles have diverse, brief experiences, such as dating multiple people simultaneously and openness to a threesome.

## 3. Commercial Casual Sex

> "I give them what they want—a hot girl to accompany them to events and no-strings-attached sex. I understand the game. They're men. They want sex. And I want their money". Maggie, a sugar baby, *Toronto Star*, 20 February 2015

After discussing various forms of casual sexual relationships and experiences, I turn to discuss commercial casual sex. I focus on the traditional phenomenon of prostitution

and, after, on the increasingly prevailing experience of sugaring, which is higher on the seriousness scale than prostitution, though perceived as lower on the normative scale.

### 3.1. Prostitution

The connection between money and sex is complex. On the one hand, people are attracted to wealthy individuals, and it is even claimed that sexual satisfaction is often greater with such people. This attraction is based on a realistic assumption that a lover will enjoy some of the benefits of this money. On the other hand, direct payment for casual sex services is commonly perceived to be base and vulgar and, in some countries, is considered a criminal offense.

A first step for coping with this puzzle is to realize the variety of commercial sex (or sex work). Christine Harcourt and Basil Donovan (2005) claim there are at least 25 types of sex work, which provide sexual services for money or its equivalent, typically stigmatized and often criminalized. Sex work takes place on the street or other public places and is probably the most typical and most widespread sex work. Less typical (in their terms, "indirect") sexual services, such as lap dancing, massages, phone sex and virtual sex, involve little or no genital contact and, therefore, have few health risks. An important variable that determines the typicality of sex work is the number of clients a sex worker sees during a typical working shift [56]. Two other related features characterizing the typicality of commercial sex work are (a) the nature and degree of intimacy—lesser intimacy is related to more typical commercial sex work—and (b) the significance of money in the experience—greater role of money relates to being closer to typical commercial sex work.

Prostitution, which is often called "the oldest profession", is typically a one-time, brief, non-intimate experience, providing immediate sexual release while money is exchanged for sex. Typical commercial sex is by far less serious than non-commercial casual sex, though repeated meetings with the same sex worker may have some similarities to booty calls. Street prostitution has an almost zero degree of seriousness since it is very brief. This is one reason for its persistence in all cultures throughout history, despite often being criminalized. However, there are types of commercial sex that involve greater degrees of seriousness. For example, some clients request the "girlfriend (or boyfriend) experience", in which sex workers provide additional activities, such as kissing, cuddling and hugging.

Turning to the moral value of commercial sex, Martha Nussbaum (1999) argues that most of us take money for the use of our body. While we believe it is right for artists and professors to use their bodies for money, it is widely believed that taking money for the use of one's sexual capacities is bad. Nussbaum claims that there is nothing baneful about taking money for a service, including intimate activities. The prostitute's activity is problematic because of features of her working conditions and the way she is treated by others. In Nussbaum's view, most committed and intimate sex can involve a contract and a financial exchange. She claims that quite frequently, committed relationships include an element of economic dependence. Marriage has frequently had this element, but it typically does not exchange money for sex [57].

In a similar vein, Jessica Flanigan (2022) defends the decriminalization of sex work, arguing that such work should be treated like other kinds of work, such as nursing, massage therapy, performance arts and marriage counseling. Criminalization does not eliminate the sex industry but makes it more risky for workers and their clients [58]. Likewise, Shulamit Almog (2019) claims that buying and selling sex is normatively neutral; in itself, these deeds are not immoral deeds and, hence, we should not fight against them [59]. Similarly, Ole Martin Moen (2014) argues that if we believe that sexual encounters need not be deeply personal and emotional in order to be acceptable, we actually accept casual sex, which refers to sex for the sake of enjoyment without long-term commitments and emotional attachments. In this case, it does not make sense to reject prostitution. This does not mean that prostitution has no downsides, but those mainly stem not in something intrinsic to prostitution but rather in contingent external factors [60].

### 3.2. Sugaring

"I didn't want a full-time partner. Instead, I fantasized about someone older, more sophisticated, more established. And, if I'm being honest, someone with some money, too . . . Given that I didn't want a committed relationship, it made sense to me". Helen Croydon, *Sugar Daddy Diaries*

As hookups, sugaring is also very popular among young people, but while hookups are extremely low on the seriousness scale, sugaring is much higher. Both seem to be of some value when they are moderate in their expressions.

Sugaring is a kind of relationship in which social and intimate needs, including sex, are provided in return for money and gifts. Thus, a "sugar baby" is someone who receives "gifts" (including cash) in exchange for company, which can include sex but does not have to. A "sugar daddy", a person who gives such "gifts", is typically wealthier and older than the sugar baby. The case of "sugar mommas" is rarer, probably since women are less willing to pay for sex. Sugaring is also present among homosexual people. Sugaring has become an increasingly common phenomenon, verging on entering the mainstream. Some types of sugaring are present in various societies throughout history. In our society, sugaring has spread to millions of people all over the world. This has caused sugaring to become almost mainstream, no longer restricted to the very rich or the very poor; those from all social and economic strata may indulge in this lifestyle.

There are various types and forms of sugaring relationships. Maren Scull (2020) identified a range of sugaring types: at one end of the continuum, there is Sugar Prostitution, and at the other end, there is Sugar Friendship and Sugar Love. Scull further claims that 40% of the women in her study who have "sugared" do not have sex with their benefactors, and these women often have genuine connections with the men. Most forms of sugaring, she notes, are not play-for-pay arrangements [61].

Like romantic love and prostitution, sugaring typically involves sex. Sugaring has behavioral elements that are typical to love and absent in prostitution. Unlike prostitution, and similar to romantic relationships, sugaring is a relationship with a certain degree of seriousness. With the exception of friends with benefits, the degree of seriousness in sugaring is higher than in other casual sexual relationships. Sugaring straddles the bridge between prostitution and serious romantic relationships, in major aspects underlying the suggested seriousness scale of intimate relationships: temporality, profundity, commitment and authenticity.

The *temporal* aspect of sugaring is more complex than in prostitution and less so than in romantic relationships. Thus, the *duration* of sugaring is longer than in prostitution, as well as most types of casual relationships, but briefer than in serious romantic relationships. The same goes for the *continuity* (or companionship) factor, referring to the time awake the couple spends together, as well as the *frequency* of their meetings. There is also a process of *meaningful development*, which is absent in prostitution; however, this process is less meaningful than that in committed romantic relationships. Sugaring also stands between prostitution and committed relationships in the issue of *profundity*. Prostitution typically involves a superficial one-time, one-dimensional, relatively brief sexual encounter, while serious romantic relationships are profound, multidimensional, ongoing interactions. Sugaring is more profound than prostitution but much less so than romantic relationships. The issues of *commitment* and trust, which are significant in romantic relationships and barely present in prostitution, exist in sugaring in a limited manner. Thus, the number of partners in a sugaring is considerably fewer than in prostitution, and the relationship is more personal and intimate. Additionally, the issue of authenticity is by far more complex in sugaring than in prostitution and most types of casual sex.

The normative challenge to traditional values posed by sugaring is greater than that of casual sex or prostitution. One reason is that the already diverse, flexible and often shaky foundations of traditional values concerning casual sex are unclear on how to approach sugaring, which lacks clear normative guiding rules. Accordingly, despite its higher level of seriousness than in prostitution, sugaring often invokes greater normative criticism and stigma.

Thus, Vojin Rakić (2020) argues that "prostitution is a categorically moral activity . . . while sugaring is hypothetically immoral . . . Unlike prostitution that should not only be legalized but also acquire the status of a morally apposite profession, sugaring is based on cheating and deserves to be outlawed, from an ethical point of view". Rakić argues that sugaring is immoral since the conditions are usually unknown to both parties in the transaction, quite different to prostitution [62]. Rakić is right that it is easier to specify the conditions of the superficial activity of prostitution than those in the complex and more serious relationship of sugaring. However, superficiality is not a sound reason for determining the moral value of human activities. Morality is all about profundity and complexity.

The negative evaluation of sugaring is not because it is too casual or because it is too serious, but because of the risk in blurring normative lines of our society. Sugaring expresses our inability to draw strict lines in the sexual realm. The contribution of sugaring to one's wellbeing is also not straightforward, and there are many circumstances in which both partners have increased wellbeing. Sugaring defies the assumed correlation between seriousness and normative value. While its degree of seriousness is typically higher than other types of casual sexual relationships, its evaluation is often more negative.

### 4. The Value of Casual Sex to Our Wellbeing

"I want to have sex all night long. Just not with my husband!" A married woman

After discussing the nature and complexity of casual sex, I turn now to examine their value to our wellbeing. This section mainly discusses recent studies suggesting the benefits of casual sex, while briefly indicating that this does not imply the absence of risks and negative consequences.

Some recent studies show that central features of casual sex are valuable to our wellbeing: (1) variety, spontaneity and a change in perspective are crucial in enriching our lives and essential for our wellbeing [63]; (2) psychological flexibility is the single most commonly proven skill of importance to our mental health and emotional wellbeing [64]; (3) superficial contact with strangers increases our wellbeing [65]; (4) people view their casual sexual encounters more positively than negatively [66]); (5) greater authenticity [1]; and (6) less cases of efficient sex and more affective behaviors in the sexual experience itself [67].

#### 4.1. The Negative Outcomes of Casual Sex

Before beginning to discuss the contribution of casual sex to our wellbeing, I would like to state the obvious: when practicing casual sex, there are various threats to our wellbeing. This section is very brief, not because the negative outcomes are negligible but rather since there are abundant studies on this issue, and I focus here on the positive outcomes of casual sex.

The prevailing view, which is supported by empirical findings, emphasizes the negative aspects of casual sex, which may lead to a host of negative, physical and psychological outcomes. Studies indicate that those who engage in casual sex may suffer emotional consequences that persist long after the details of an encounter are a dim memory. Those who engage in more casual sexual encounters may have greater psychological distress and regret, as well as greater tendencies toward drinking and drug problems [68–72]. It was also found that Tinder users take more health/safety risks and have a lower level of sexual disgust than non-users [73]. Additionally, staying married in midlife is associated with a lower risk of dementia; divorcees account for a substantial share of dementia cases [74].

I agree that casual sex may have harmful consequences that people should try to avoid. However, in light of opposite findings, it is clear that some people can enjoy casual sex without negative consequences. Even if we accept the above negative findings as genuine, this will not contradict my main claim indicating the benefits of casual sex to various people in certain circumstances. The possible threats are potential in two senses: (a) they refer to people who may, but do not have to, suffer from actualization of the threats, and (b) there are various personal and behavioral manners that may reduce, or even eliminate, these

threats. Thus, conducting safe behavior may significantly reduce the costs of casual sex. There are many risks in our life, and we are not supposed to stop living because of them. Rather, we are supposed to live in a more careful, and typically balanced, manner that decreases the threats and enhances the benefits.

I turn now to examine the various benefits that casual sex may have.

### 4.2. Richness in a Good Life

Happiness and meaningfulness are frequently mentioned when characterizing wellbeing. Shgehiro Oishi and Erin Westgate (2021) add another dimension: psychological richness. They claim that whereas a happy life is characterized by comfort, joy and stability, and a meaningful life by purpose, significance and coherence, a psychologically rich life is portrayed by variety, interesting experiences and changes in perspective. Oishi and Westgate argue that stable relationships, time, money and positive mindsets facilitate a happy life; strong moral principles and religiosity facilitate a meaningful life; and curiosity, spontaneity and energy enhance a psychologically rich life. They suggest that since openness to new experiences and curiosity encourage individuals to pursue and appreciate novel, complex, challenging, potentially perspective-changing experiences, and willingness to defy traditional attitudes, they constitute dispositional factors that facilitate a psychologically rich life [63].

How do these dimensions of wellbeing relate to intimate relationships? The major features of meaningfulness, i.e., significance, coherence and purpose, are indeed more dominant in enduring, serious romantic relationships. Happiness, characterized by comfort, joy and stability, is also more typical in serious romantic relationships, though shortterm intense joy can be found more in casual sex. A psychologically rich life, which involves variety, interesting experiences, a change in perspective, curiosity and openness, is closer to casual sex than to serious romantic relationships. Serious relationships can be psychologically rich, for they include a wide variety of experiences and a change in perspective, such as taking into account one's partner's perspective. However, the degree of richness in casual sex is typically greater.

There are differences in the richness of casual sex, which are often negatively correlated with the seriousness of the relationship. Thus, one-night stands are typically psychologically richer than friends with benefits, which are richer than committed relationships. When marriages take the form of consensual nonmonogamy, such as in open marriages or polyamory, these relationships are typically richer than monogamous marriages [75,76].

### 4.3. Psychological Flexibility

"Good girls go to heaven; bad girls go everywhere". Mae West, American actress

Steven Hayes and colleagues (2022) argue that psychological flexibility is the single most commonly proven skill of importance to our mental health and emotional wellbeing. The three pillars of psychological flexibility are (a) awareness—noticing what happens in the present moment; (b) openness—dropping the internal fight, allowing thoughts and feelings to be what they are without them needing to control us; and (c) valued engagement—knowing what matters to you and taking steps in that direction [64].

Psychological flexibility relates to psychological richness; both involve awareness and openness that are typically essential parts of casual sex. In both richness and flexibility, change and novelty are central. One difference between the two studies is that the study on richness focuses on the good life, whereas the study on flexibility mainly addresses mental health. Psychological flexibility is much greater in casual sex than in serious romantic relationships. This is particularly true concerning openness in the sense of lesser control, but also regarding awareness and various valued engagement—even if the value is limited. Thus, studies on wisdom show that wisdom predicts greater wellbeing and that openness is the personality trait most strongly linked to wisdom [77]. It is clear that openness is dominant not merely in wisdom but in casual sex as well.

### 4.4. Social Contact with Strangers

"I think we can all agree that sleeping around is a great way to meet people".
Chelsea Handler, American comedian

Joe Keohane (2021) argues that coping with unfamiliar outsiders not only civilizes us but might be the key to our survival and thriving [78]. Similarly, Paul Van Lange and Simon Columbus (2021) claim that our wellbeing is not merely served by the quality of close relationships but also through social contacts with people whom we know less well, even strangers. They show that most strangers are benign, and most interactions with them are positive and enhance our wellbeing. They further claim that situations with strangers often represent a low conflict of interest and that in interactions with strangers, most people exhibit minimal efforts, and if the need is urgent, more effort is available [65].

As opposed to enduring profound love, casual sex reaps more of the benefits that strangers can provide. As Russell Vannoy claims, when having sex with a stranger, one does not always know what to expect and this lends a sense of adventure and excitement in the act, enabling partners to discover new possibilities for heightening sexual pleasure [79] (p. 123). Moreover, the excitement of sex with new partners typically increases sexual intensity.

As Wednesday Martin [80] quipped, monogamy sounds like monotony, and while we may judge an adulterous woman harshly, we have to admit she is anything but boring.

### 4.5. Subjective Evaluations of Casual Sexual Encounters

Rose Wesche and colleagues (2021) found that people evaluate their casual sexual encounters more positively than negatively. These encounters have often been associated with short-term declines in emotional health, though there is little evidence that they are detrimental in the long term. They further found that women and individuals with less permissive attitudes toward casual sex experienced worse emotional outcomes as a result. Those familiar with their casual sex partner generally have more positive emotional outcomes. If the encounter involved penetrative contact, it was more likely to be a negative experience [66,81].

### 4.6. Authentic Motivation

The most significant justification for casual sex is the reduced sexual desire over time, while being with the same partner. This is the "Coolidge Effect", which was coined for the phenomenon in which males (and to lesser extent females) in mammalian species exhibit renewed sexual interest when introduced to new sexual partners [82]. Indeed, Gurit Birnbaum (2018) claims that sexual desire tends to be strong during the early stages of a romantic relationship before subsiding gradually, with many couples failing to maintain sexual desire in their long-term relationships. However, she also claims that desire is not inevitably doomed to die with the passing of time, and not everyone will eventually lose sexual interest in each other. Since sexual desire contributes most at the earlier stages of the relationship, the intensity of sexual desire by itself cannot predict the success of long-term relationships [83].

The Coolidge effect indicates that having casual sex with a new partner is an authentic desire, with a robust biological basis. Zhana Vrangalova and Anthony Ong (2014) argue that acting authentically, in congruence with one's desires and values, has been found in many studies as promoting health and thriving. Acting inauthentically, on the other hand, is detrimental to wellbeing. They argue that such authenticity concerns casual sex as well. This is particularly true concerning those high in sociosexuality, which reflects one's willingness to engage in non-committed sexual relationships. For those people, casual sex represents an authentic and self-congruent pursuit and, hence, will be beneficial for them. Indeed, sociosexually unrestricted students typically reported higher wellbeing after having casual sex compared to not having had casual sex [1]. Another study found that feelings of authenticity are strongly associated with many facets of wellbeing and negatively associated with depression and anxiety [84].

John Townsend and colleagues (2020) found that the ability to avoid these negative consequences may be a function of people's motives. Thus, when people's motives for casual sex involved autonomous (intrinsic) motives concerning romance or sexual pleasure, their experiences were associated with more positive outcomes. Conversely, people reported more adverse psychological effects when their motives were non-autonomous (extrinsic) ones (e.g., they wanted to please someone else, gain a favor or feel better about themselves) [85]. Likewise, Val Wongsomboon and colleagues (2022) found that as well as greater sexual assertiveness, women's autonomous motivations to have casual sex are associated with higher orgasmic function, and, consequently, higher wellbeing, whereas nonautonomous motivation is associated with lower orgasmic function in casual sex, thereby decreasing their wellbeing [86].

Needless to say, authenticity can be part of an enduring committed relationship. However, my major claim is not that serious romantic relationships are not valuable—they certainly are; the claim is rather that casual sex has its own unique benefits.

### 4.7. Affective Behavior and Efficient Sex

"Last night I had sex with my husband, but he did not actually touch me—just penetrated me. I was so sad, I could cry". A married woman

Casual sexual encounters are briefer than romantic relationships, but the duration of the sexual experience itself, involving both foreplay and intercourse, is usually longer. This enables such sexual encounters to avoid "efficient sex" and to include more affective behaviors. The phenomenon of efficient sex, or "quickies", is common in long-term romantic relationships. Efficient sex is fast: once you know how to give your partner pleasure and reach orgasm, you can then fall into a routine of getting straight to those same motions. It was found that most men and women long for the non-efficient sex in which the sexual experience (both intercourse and foreplay) lasts longer [87,88]. In casual sex, efficient sex is usually less common. Aside from certain forms of hookups and commercial street prostitution, efficient sex is not the norm in the types of casual sex discussed here—one-night stands, booty calls, fuck buddies, friends with benefits and sugaring. Thus, whereas one-night stands are brief relationships (merely one night), they tend to have a lengthy sexual experience (for the entire night). Likewise, the aforementioned journalist, Lee-Zogbessou, claimed that the key difference between her previous partners and her fuck buddy was prolonged foreplay.

Indeed, in the study of Wesche and colleagues [67], in which most participants said that their casual sex experience was more positive than negative, other affective benefits were reported, such as experiencing sexual pleasure, having fun and boosting self-esteem. Moreover, Justin Garcia and colleagues (2018) found that emerging adults (aged 18–25 years) who preferred casual sex encounters (compared to romantic relationships) were more likely to engage in intimacy-motivated affectionate behaviors (such as cuddling, spending the night and cuddling, foreplay and eye gazing) during sex in the context of casual sex encounters. This was found to be more so than from participants who preferred romantic relationships [70].

In summary, casual sex takes various forms and many of them involve lengthy affective behavior, certainly not efficient sex.

### 5. Concluding Remarks: The Balance of the Good Life

The answer to our main question, "Is casual sex good for you?", can be **YES**. However, it depends on the given circumstances and individual differences, such as sociosexuality, the nature of motivation and the level of attachment security. Thus, for people with high sociosexuality, casual sex enhances their wellbeing, while for those suffering high anxiety, casual sex will likely damage their wellbeing. Such discrepancies may increase or decrease in different types of casual sex, whereas others may be less sensitive to the different types. Alicia Segovia and colleagues (2019) found that while anxious individuals' physical pleasure and positive/negative emotions are largely generalized across all types of casual sexual encounters, avoidant individuals experience different levels of physical

pleasure and positive emotions across different casual sexual encounters. Thus, avoidant individuals report the highest levels of physical pleasure and positive emotions in fuck buddy encounters [89,90].

The issue for most types of personalities is not whether to stop or even reduce casual sex but how to conduct it, when one wants it, while still lessening its risks and enhancing its benefits. There are many ways of achieving this but the common thread of the approach is through balance. Flourishing intimate relationships and experiences need both seriousness and casualness. The nature and the extent of each depend on personal and circumstantial factors but should typically be restricted in a way that does not eliminate the other aspect. Indeed, among the casual sexual relationships and experiences discussed here, the most criticized ones were the popular hookup culture and sugaring, both of which present significant risks to the subtle balance between casualness and seriousness in intimate encounters. Hookup culture challenges the normative assumption that some degree of seriousness should be present in all noncommercial sexual encounters, while sugaring shakes the assumption that commercial sexual encounters can involve such a high degree of seriousness.

It is often assumed that only enduring serious romantic relationships can reach genuine high levels of wellbeing, while casual sex can merely enhance some lower-level aspects of our wellbeing. I have shown that this assumption is incorrect and casual sex can provide significant elements of the good life (eudaimonia), and, in some aspects, can do so more than serious romantic relationships. The difficulties in having the good meaningful life in the romantic realm often stem from a lack of balance between casual and serious intimate activities. In Aristotle's view, a person is bad by virtue of lacking proper balance, not by virtue of pursuing necessary pleasures, such as dainty foods, wines and sexual intercourse. Aristotle considered not only emotional excess to be harmful but also an emotional lack. Balance does not always mean moderation, since in some extreme circumstances, our reactions should not be moderate. Indeed, in moments of extreme danger, one's reactions may need to be extreme. Similarly, with younger people, the appropriate romantic attitudes might be those of greater intensity.

The fact that we wish to thrive over time does not mean that we cannot enjoy the moment. After all, we live in the present and it is typically worthwhile to make each moment as pleasurable and meaningful as possible. However, to give priority to the moment over lasting flourishing is to neglect other key dimensions in the good life. We do not merely live in the present but we are shaped by the past and dream about the future.

I do not advocate the proverb "Eat, drink, and be merry, for tomorrow we die", but I do not think that we should be on harsh diets throughout all of our lives. A balanced intimate diet is what we need for a good and healthy life. There are various exciting ways to maintain such a diet, many of which may include some kind of casual sex.

**Funding:** This research received no external funding.

**Institutional Review Board Statement:** Not applicable.

**Informed Consent Statement:** Not applicable.

**Data Availability Statement:** Not applicable.

**Acknowledgments:** I am grateful to Raja Halwani for his insightful comments that considerably improved the quality of the article.

**Conflicts of Interest:** The author declares no conflict of interest.

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
