# Peer review of "Is Casual Sex Good for You? Casualness, Seriousness and Wellbeing in Intimate Relationships"

_philosophies, doi:10.3390/philosophies8020025_

Round 1

Reviewer 1 Report

This paper's central thesis is that without prejudice to the value of serious long-term romantic relationships, much evidence supports the claim that  "intimate relationships which have a low degree of seriousness, such as those involved in CSREs, may enjoy benefits regarding our wellbeing that those higher on the seriousness scale may not have. "   Appropriately enough, there are a number of references to empirical studies. Consequently, while the paper is somewhat slight in philosophical content, it makes a good case for its central thesis. It is readable and interesting.

A couple of small problems:

·       Some empirical claims are devoid of citations.  Two examples:  "People often count the frequency of their sexual encounters and the number of orgasms in each" ;  "Profound activities, which are sometimes unpleasant, typically involves deep, meaningful satisfaction in overcoming difficulties while using one’s most distinctive capacities." The latter quote is embedded in a discussion of profundity, intensity and commitment in romantic relationships. These are familiar themes in some of the work of Ben Ze'ev, by whom five papers are listed but without specific citations in that passage. Specific claims should be supported by specific citations.

·       “Another problematic aspect of prostitution is that it is a trade that people do not enter by choice. (Nussbaum 1999)" (p. 11). Nussbaum's paper does support that unqualified generalization.  She writes that "many of women’s employment choices are so heavily constrained by poor options that they are hardly choices at all.... the fact that a woman with plenty of choices becomes a prostitute should not bother us, provided that there are sufficient safeguards against abuse and disease, safeguards of a type that legalization would make possible." So what Nussbaum writes is quite compatible with the fact that some persons do enter that profession by choice.

·       Is ‘Causality’ the right word in the title? Casualness is what is discussed in the paper.

·       Yam, F. C. (2022).   Reference is out of order in the bibliography

Author Response

I would like to thank the editor and the reviewers for their most insightful and useful comments. I have accepted almost all of these comments and have considerably revised the text.

I have revised the text in light of the comments of Reviewer 1 (including the suggestion regarding Nussbaum’s view).

Reviewer 2 Report

This is a nicely written defense of casual sex. It's well-informed and well-written, and deserves to be published in Philosophies. I do, however, suggest an even more confrontational title: "A Defense of Casual Sex".

Author Response

I would like to thank the editor and the reviewers for their most insightful and useful comments. I have accepted almost all of these comments and have considerably revised the text.

Reviewer 2 suggests changing the title to "A Defense of Casual Sex", which the reviewer believes to be more provocative. The suggested title may indeed be more provocative, but after some consideration, I would prefer to keep the original title, which I believe more adequately expresses the spirit of the article.